Status of the undisturbed mangroves at Brunei Bay, East Malaysia: a preliminary assessment based on remote sensing and ground-truth observations

Satyanarayana Behara satyam@umt.edu.my satyam2149@gmail.com 1 2
M. Muslim Aidy aidy@umt.edu.my 1
Izzaty Horsali Nurul Amira 1
Mat Zauki Nurul Ashikin 1
Otero Viviana 2
Nadzri Muhammad Izuan 1
Ibrahim Sulong 1
Husain Mohd-Lokman 1
Dahdouh-Guebas Farid 2 3
1 Universiti Malaysia Terengganu—UMT, Mangrove Research Unit (MARU), Institute of Oceanography and Environment (INOS) , Kuala Terengganu , Malaysia
2 Université Libre de Bruxelles—ULB, Laboratory of Systems Ecology and Resource Management , Brussels , Belgium
3 Laboratory of Plant Biology and Nature Management, Vrije Universiteit Brussel , Brussels , Belgium
Minasny Budiman
Electronic publication date: 2018 Feb 20
Publication date: 2018
Volume: 6
Electronic Location ID: e4397
Received 2017 Sep 13; Accepted 2018 Jan 30
Copyright: ©2018 Satyanarayana et al.
Copyright year: 2018
Copyright holder: Satyanarayana et al.
License: This is an open access article distributed under the terms of the Creative Commons Attribution License, which permits unrestricted use, distribution, reproduction and adaptation in any medium and for any purpose provided that it is properly attributed. For attribution, the original author(s), title, publication source (PeerJ) and either DOI or URL of the article must be cited.
License URL: https://creativecommons.org/licenses/by/4.0/

Keywords: ALOS data, Diameter class distribution, Species-level mapping, PCQM, South China Sea

Funding: INOS, UMT The present research work was supported by the HiCOE program at the INOS, UMT. The funders had no role in study design, data collection and analysis, decision to publish, or preparation of the manuscript.

==============================
Brunei Bay, which receives freshwater discharge from four major rivers, namely Limbang, Sundar, Weston and Menumbok, hosts a luxuriant mangrove cover in East Malaysia. However, this relatively undisturbed mangrove forest has been less scientifically explored, especially in terms of vegetation structure, ecosystem services and functioning, and land-use/cover changes. In the present study, mangrove areal extent together with species composition and distribution at the four notified estuaries was evaluated through remote sensing (Advanced Land Observation Satellite—ALOS) and ground-truth (Point-Centred Quarter Method—PCQM) observations. As of 2010, the total mangrove cover was found to be ca. 35,183.74 ha, of which Weston and Menumbok occupied more than two-folds (58%), followed by Sundar (27%) and Limbang (15%). The medium resolution ALOS data were efficient for mapping dominant mangrove species such as Nypa fruticans, Rhizophora apiculata, Sonneratia caseolaris, S. alba and Xylocarpus granatum in the vicinity (accuracy: 80%). The PCQM estimates found a higher basal area at Limbang and Menumbok—suggestive of more mature vegetation, compared to Sundar and Weston. Mangrove stand structural complexity (derived from the complexity index) was also high in the order of Limbang > Menumbok > Sundar > Weston and supporting the perspective of less/undisturbed vegetation at two former locations. Both remote sensing and ground-truth observations have complementarily represented the distribution of Sonneratia spp. as pioneer vegetation at shallow river mouths, N. fruticans in the areas of strong freshwater discharge, R. apiculata in the areas of strong neritic incursion and X. granatum at interior/elevated grounds. The results from this study would be able to serve as strong baseline data for future mangrove investigations at Brunei Bay, including for monitoring and management purposes locally at present.

Introduction

A combination of ground truth and remote sensing data analysis is advantageous for developing the most reliable land-use/cover mapping and thereby useful for making appropriate decisions for conservation and management of natural resources (Kovacs, Wang & Blanco-Correa, 2001; Kovacs et al., 2004; Chauhan & Dwivedi, 2008; Neukermans et al., 2008; Satyanarayana et al., 2011; Leempoel et al., 2013). In the case of mangroves, remote sensing data have become indispensable due to its time saving and cost-effective nature compensating for the fieldwork, which is often difficult to carry out, especially in areas of low accessibility (Giri et al., 2007; Dahdouh-Guebas & Koedam, 2008; Giri et al., 2008; Massó i Alemán et al., 2010; Giri et al., 2011; Satyanarayana et al., 2011; Cárdenas, Joyce & Maier, 2017). Mangrove mapping—as per the target of identifying different vegetation details—has been conducted with low to very high-resolution spaceborne (e.g., Landsat, IKONOS, QuickBird, GeoEye-1) and airborne remote sensing data (Dahdouh-Guebas et al., 2000; Sulong et al., 2002; Wang et al., 2004; Seto & Fragkias, 2007; Giri et al., 2008; Dahdouh-Guebas & Koedam, 2008; Spalding, Kainuma & Collins, 2010; Giri et al., 2011; Hansen & Loveland, 2012; Leempoel et al., 2013; Proisy et al., 2016). Also, the potential of moderate resolution data like Advanced Land Observation Satellite (ALOS) and Sentinel for mangrove studies is well recognised (Hartoko et al., 2015; Castillo et al., 2017; Chen et al., 2017). The optical remote sensing data which were often limited by cloud cover to study mangrove ecosystems have been compensated through the radar and drone imageries in recent years (Walters et al., 2008; Cornforth et al., 2013; De Santiago, Kovacs & Lafrance, 2013; Kovacs et al., 2013; Hamdan, Khali Aziz & Mohd Hasmadi, 2014; Lucas et al., 2014; Jhonnerie et al., 2015; Aslan et al., 2016; Tian et al., 2017). Similarly, there are several plot-based and plot less methods which exist for undertaking vegetation inventories (Elzinga et al., 2001). Among others, the Point-Centred Quarter Method (PCQM) is recognised as the most appropriate ground-truth for mangrove and remote sensing combinatory investigations (Cintrón & Schaeffer Novelli, 1984; Dahdouh-Guebas & Koedam, 2006; Satyanarayana et al., 2011). The PCQM is not only efficient for characterising mangrove vegetation and less time-consuming, but it also causes minimum damage to the understorey while sampling (Cunningham, 2001; Dahdouh-Guebas & Koedam, 2006).

The mangrove forest in Malaysia (709,700.00 ha) is the second largest in Southeast Asia and sixth (after Indonesia, Brazil, Australia, Mexico and Nigeria) among the nations that supporting highest mangrove cover in the world (Spalding, Kainuma & Collins, 2010; Hamdan et al., 2012). East Malaysia (i.e., Sabah and Sarawak) is supporting up to 84% of the country’s mangroves and West (Peninsular) Malaysia the remaining 16% (Table 1). Mangroves at Brunei Bay are jointly shared, but separately administered by East Malaysia and Brunei Darussalam (Fig. 1). Although known as one of the largest tracts of relatively undisturbed forest in eastern Asia (Brunei Darussalam Forestry Department, 2010), the mangroves at Brunei Bay have been less scientifically explored and much of the information on vegetation structure, ecosystem services and functioning, and land-use/cover changes is confined to grey literature (e.g., Justin, 2007; Ali & Mohd Ariff, 2007; Malaysian Ministry of Natural Resources and Environment, 2014). According to Adiana et al. (2017), there were only five scientific publications between 1968 and 2012 on Brunei Bay, and they related to water pollution and marine mammals in the area. The research works after 2012 (e.g., Alkhadher et al., 2015; Joseph et al., 2016; Proum et al., 2016; Joseph et al., 2017) also focused on sediment pollution and sea turtles, but none were specific to the (Malaysian) mangrove vegetation.

Table 1 Mangrove cover in Malaysia.

State-wise distribution of mangrove cover in West (Peninsular) and East Malaysia (source: Hamdan et al., 2012).

	State	Mangrove cover (ha)	% contribution	
1. Peninsular (West) Malaysia				
West coast:	Perak	43,291.97		
	Johor	23,676.43		
	Selangor	22,530.20		
	Kedah	7,841.25		
	Negari Sembilan	2,276.50		
	Pulau Pinang	1,695.60		
	Melaka	1,308.68		
	Perlis	94.02		
Total West coast:	–	100,438.15	14.15	
East coast:	Pahang	9,039.26		
	Terengganu	2,925.74		
	Kelantan	428.95		
Total East coast:	–	12,393.95	1.75	
2. East Malaysia	Sabah	426,334.20		
	Sarawak	170,533.70		
Total East Malaysia	–	596,867.90	84.10	
	Total	709,700.00	100.00	

Figure 1 Study area location.

Location of Brunei Bay in the East Malaysia. The mangrove floristic surveillance was carried out from four major estuarine areas namely, Limbang (box 1), Sundar (box 2), Weston (box 3), and Menumbok (box 4). The town/village names in the vicinity of Brunei Bay were represented by black bullets. The area under the jurisdiction of Brunei Darussalam was masked in yellow colour (ALOS satellite image dated 1 Sept 2010). The white triangle named after MM on the satellite image shows the location of missing mangrove patch on the northernmost corner of Menumbok estuary.

The present study was primarily aimed at identifying the current status of the Malaysian mangrove cover at Brunei Bay. The objectives were to develop a species-level classification map based on remote sensing (ALOS) data for Limbang, Sundar, Weston and Menumbok estuaries adjoining Brunei Bay, and to integrate and validate the findings through ground-truth (PCQM) observations.

Materials and Methods

Study area

Brunei Bay is extended over 2,500 km2 where much of its aquatic and terrestrial land belongs to East Malaysia (Fig. 1). The bay area is also known as a significant habitat for marine biological diversity in the South China Sea (Vo, Pernetta & Paterson, 2013; Joseph et al., 2016). The southern limit of the bay is flanked by lush green mangroves, especially on the lower reaches of Limbang, Sundar, Weston and Menumbok rivers (BirdLife International, 2015). The Limbang estuary—emerging from the three rivers, namely Sungai Manunggul, Sungai Limbang and Sungai Pandaruan—is orientated in a north-south direction whereas the Sundar estuary—emerging from Sungai Trusan—is stretched between Kuala Trusan and Tanjung Perepat in a west-east direction. Both Weston and Menumbok estuaries are formed by Sungai Padas and Sungai Klias, respectively, and run in a north-east direction. The mangrove vegetation, together with seagrass beds and coral reefs in Brunei Bay, are providing several eco-socio-economic benefits to the local communities (Ahmad-Kamil et al., 2013). This study was conducted with the permission of the State Forestry Department of Sarawak (# NCCD.907.4.4 (Jld. 10)-294).

The climate of Brunei Bay is influenced by tropical weather with two monsoonal regimes: northeast (mid-December to mid-March) and southwest (mid-May to the end of October) (Malik, 2011). The historical weather data (2005–2015) showed an average highest temperature of 33.5 °C for April–May and the highest precipitation of 72.9 mm for June (WU, 2015). Intense seawater current, accompanied by strong winds, can be observed every year during the northeast monsoon (Nelson et al., 2015). Monsoonal impact in the areas facing the South China Sea is evident by coastal erosion and in less and patchy distribution of the mangrove cover (Hamdan et al., 2012).

Remote sensing data and analysis

For mangrove species-level mapping at Brunei Bay, the Advanced Visible and Near-Infrared Radiometer type 2 (AVNIR-2) data acquired from the ALOS (spatial resolution: 10 m) (dated 1st September 2010) were used. The ALOS data—subjected to both atmospheric and geometric corrections—were provided by the Japan Science and Technology Agency (JST). However, to ensure a good match of the land-use/cover features in the imagery with the ground-truth observations, the data have been georeferenced again with WGS_1984 coordinate system using a toposheet (1:50,000) obtained from the Department of Survey and Mapping Malaysia (RMS error: 0.682) (ArcMap v.10). In order to have a better image processing and analysis, the mangrove areas adjacent to the bay and river channels were digitised on screen. For the medium resolution remote sensing data like ALOS, on screen digitisation through visual interpretation is indeed beneficial to separate mangrove and non-mangrove areas (Kuenzer et al., 2011). In this context, the false colour composite (FCC) (with 4-3-2 band combination) of the ALOS data was used to recognise the mangroves visible in a darker shade than of nearby terrestrial vegetation due to less spectral reflectance (Spalding, Kainuma & Collins, 2010; Zhang et al., 2017). The polygon features, as a new shapefile (with WGS_1984 coordinate system), were created only for mangroves under the jurisdiction of East Malaysia, and extracted the raster cells using spatial analyst tools (with ‘extract by polygon’ function in ArcMap v.10; ArcGIS, Redlands, CA, USA).

Species-level classification using all four (blue, green, red and near-infrared) bands was carried out through the maximum-likelihood algorithm that known to facilitate a robust classification for mangroves (Wang, Sousa & Gong, 2004; Shafri, Suhaili & Mansor, 2007; Satyanarayana et al., 2011; Kuenzer et al., 2011; Nguyen et al., 2013; Khatami, Mountrakis & Stehman, 2016; Mafi-Gholami, Mahmoudi & Zenner, 2017). In this context, the training samples for the most dominant mangrove taxa such as Nypa, Rhizophora, Sonneratia and Xylocarpus spp., were assigned based on ground knowledge acquired through the PCQM (68 sample points: Limbang—9, Sundar—19, Weston—20 and Menumbok—20) (details are given in ‘Ground inventory’). In addition, tonality and textural characteristics (in FCC) of the dominant species were considered (cf. Dahdouh-Guebas et al., 2000; NRC, 2006). For instance, Nypa vegetation was represented by a dark red colour with coarse texture, whereas Rhizophora by bright red with fine texture, Sonneratia by light red with open spaces (especially at river mouths), and Xylocarpus by a bright red colour with coarse texture (especially at back mangrove area). Based on our ground knowledge and visual interpretation of the satellite data, we assigned 20–25 training samples (with a pixel count of 125,135–572,692) for the widespread mangrove species (Nypa, Rhizophora spp.), and 10–15 (with a pixel count of 98,610–102,029) for the locally distributed species (Sonneratia, Xylocarpus spp.). The classified image was then subjected to an accuracy assessment through a confusion matrix (Congalton, 1991), for which 144 additional ground control points (GCPs) (Limbang—60, Sundar—45, Weston—26 and Menumbok—13) collected from both mangrove and non-mangrove areas were used. The GCPs were collected randomly as per the forest condition and accessibility. The location of the GCPs was marked on the ALOS imagery (hardcopy) and recorded the type of land-use/cover (i.e., mangrove or non-mangrove), together with species composition in the case of mangrove, for the accuracy assessment. Most locations in mangrove from where the GCPs were collected have shown the distribution of dominant species in more than 10 m × 10 m land-cover area.

For the accuracy assessment, we also report quantity (%)—the amount of pixels that differed between reference data and classification per class, exchange (%)—the allocated error by number of pixels that interchanged between two classes, and shift (%)—the other allocation differences that were not included in the exchange difference (cf. Pontius Jr & Millones, 2011; Pontius Jr & Santacruz, 2014). Any disagreements among quantity, exchange and shift variables are useful to learn the sources of error in the classification in a more interpretable manner (Pontius Jr & Millones, 2011). Estimates of quantity, exchange and shift were made from the PontiusMatrix41 (Microsoft Excel file) developed by RG Pontius (https://www.clarku.edu/).

Area statistics showing the total mangrove cover (with a species-level demarcation for each estuary) at Brunei Bay were derived from the supervised classification. For this purpose, the mangrove extent was treated under three sectors: Limbang, Sundar (up to Sipitang village) and Weston + Menumbok together (due to no clear-cut mangrove boundary between these two estuaries). However, there were a few limitations identified with the ALOS data. Firstly, the image was four years old at the time of fieldwork. Second, a (minor) patch of mangrove was missing on the northern most corner of Menumbok (Fig. 1). The former concern was evaluated through our ground inventory by observing the mangrove cover changes (if any), while the latter was identified from Google Earth Pro (image dated 29th November 2014) by digitising the missing (mangrove) area.

Ground inventory

For the ground data collection (5th–19th August 2014), two 100 m length transects—one close to the river mouth and another in mid-forest—were chosen from each estuary i.e., Limbang, Sundar, Weston and Menumbok (eight transects in total). Sampling at the river mouth was chosen to identify the pioneer group of vegetation (i.e., mangrove succession by select species), whereas in the mid-forest to identify other available species in the vicinity. We took a 100 m transect in order to cover at least ten sample points with four quadrants of the PCQM (cf. Engeman et al., 1994; Satyanarayana et al., 2002; Satyanarayana et al., 2011). Different adult tree (i.e., ≥1.3 m height with a diameter (D130) of ≥2.5 cm or girth (G130) ≥8 cm) structural parameters such as density (trees 0.1 ha−1), basal area (m2 0.1 ha−1), relative density (%), relative dominance (%), relative frequency (%), species’ importance value (IV) (relative density + relative dominance + relative frequency), and the complexity index (CI) were estimated using the P-DATA PRO v. 5.01 interface developed by Dahdouh-Guebas & Koedam (2006). In the case of Nypa—as its stem remains underground—the diameter of all leaf shoots was considered to determine the (average) basal area. For mangrove species identification, the nomenclature suggested by Tomlinson (1986) and Duke (2006) was followed. Each tree height was measured with the help of a clinometer (Suunto PM-5, Finland). A hand-held Global Positioning System (Garmin 45; Garmin, Olathe, KS, USA) was used for navigation and to obtain the latitude and longitude positions of the sampling points.

Statistical analysis

Variation between the tree structural parameters, like density and basal area, at Limbang, Sundar, Weston and Menumbok estuaries was tested through one-way ANOVA (OriginPro v. 9.1).

Figure 2 Mangrove supervised classification.

(A) Supervised classification of the mangrove vegetation at Brunei Bay. The species-level distribution of mangroves (white boxes) at (B) Limbang, (C) Sundar and, (D) Weston + Menumbok estuaries The arrows named after L1, L2 in (B), S1, S2 in (C) and, W1, W2, M1 and M2 in (D) shows the vegetation survey (PCQM) sampling points at those respective estuarine areas (ALOS single band satellite image dated 1 Sept 2010) (the mangrove area under the jurisdiction of Brunei Darussalam was ignored from image processing/analysis, see Fig. 1 for country’s boundary). The white box in (D) shows digitized mangrove cover (from the Google Earth Pro image dated 29 Nov 2014) on the northernmost corner of Menumbok estuary.

Results

Remote sensing based observations

From the supervised classification of the ALOS data (Fig. 2), it was possible to recognise the predominance of Sonneratia caseolaris (L.) Engler, Rhizophora apiculata Bl. and Nypa fruticans (Thunb.) Wurmb., along with S. alba J Smith and Xylocarpus granatum König species at Brunei Bay (accuracy: 80% and Kappa index: 0.714) (Table 2). The difference in the amount of pixels between the reference data and the classification per class (= quantity) was found to be 9%, whereas the error due to interchanged pixels between two classes (= exchange) was 7% and the allocation difference other than to the exchange difference (= shift) was 1%. Among the five mangrove species, N. fruticans showed the highest quantity (9%) and exchange (7%) differences (Fig. 3). In fact, there was a considerable overlap in the (visible range) spectral reflectance values of the dominant mangrove species (Fig. 4). The Malaysian mangrove cover at Brunei Bay was found to be ca. 35,183.74 ha of which Limbang occupied 5,011.42 ha, Sundar (up to Sipitang village) 9,606.46 ha, and Weston + Menumbok 20,565.86 ha (Table 3). If the spatial extent of each mangrove species is considered, N. fruticans shows a widespread distribution (14,879.27 ha), followed by R. apiculata (12,801.89 ha), S. caseolaris (5,533.10 ha), X. granatum (993.10 ha) and S. alba (976.38 ha) (Table 3). Area wise, Limbang, Weston and Menumbok were dominated by N. fruticans (as an important species), and Sundar by R. apiculata (Table 3).

Table 2 Accuracy assessment of the supervised classification.

Confusion matrix showing the accuracy assessment of species-level mangrove supervised classification at Brunei Bay.

		Supervised classification	
		R. apiculata	S. caseolaris	S. alba	N. fruticans	X. granatum	Total	Producer’s accuracy (%)	
Ground-truth	R. apiculata	46	1	2	1	0	50	92	
S. caseolaris	0	12	2	1	0	15	80	
S. alba	0	0	12	0	0	12	100	
N. fruticans	11	6	1	23	0	41	56	
X. granatum	0	0	0	0	5	5	100	
	Total	57	19	17	25	5	123		
User’s accuracy (%)	81	63	71	92	100			
Notes.

Genus names R Rhizophora

S Sonneratia

N Nypa

X Xylocarpus

Figure 3 Pontius matrix.

Quantity, Exchange and Shift differences in species-level mangrove supervised classification at Brunei Bay (Genus names: R, Rhizophora; S, Sonneratia; N, Nypa and X, Xylocarpus). The X-axis refers to the difference (%) in each mangrove category of the study area.

Figure 4 Spectral reflectance of mangrove and non-mangrove vegetation.

Average spectral reflectance curves of the dominant mangrove species and the adjacent terrestrial vegetation at Brunei Bay.

Ground inventory observations

Ground-truth observations found several non-dominant mangrove species like Avicennia alba Blume, Bruguiera gymnorrhiza (L.) Lamk., B. cylindrica (L.) Blume, Ceriops sp., Heritiera littoralis Dryand and Kandelia candel (L.) Druce, along with the associates such as Acanthus ilicifolius L., Acrostichum aureum L., Derris trifoliata Lour. and Hibiscus tilliaceus L., in the vicinity. Out of 11 dominant and non-dominant mangrove species, only nine were encountered in the vegetation (PCQM) sampling points (Table 4). Among the four estuaries, Limbang had the highest mangrove basal area (120.17 m2 0.1 ha−1 with a density of 163 trees 0.1 ha−1), followed by Menumbok (81.17 m2 0.1 ha−1 with a density of 132 trees 0.1 ha−1), Weston (35.10 m2 0.1 ha−1 with a density of 130 trees 0.1 ha−1), and Sundar (33.24 m2 0.1 ha−1 with a density of 134 trees 0.1 ha−1). Basal area at Limbang and Menumbok was significantly different from Sundar and Weston (one-way ANOVA, P = 0.08). Sonneratia caseolaris holds the highest importance value for Limbang, while R. apiculata for both Sundar and Menumbok, and N. fruticans for Weston. The mangrove stand structural complexity (derived from the complexity index) was high in the order of Limbang >  Menumbok >  Sundar >  Weston (Table 4). In terms of the diameter class distribution, more than 50% of trees at Limbang were represented by 31–90 cm (range: 9.5–198.9 cm), Sundar 2.5–40 cm (range: 3.2–186.1 cm), Weston 2.5–60 cm (range: 3.0–190.1 cm), and Menumbok 2.5–120 cm (range: 2.5–250 cm) (Fig. 5).

Table 3 Mangrove area.

Mangrove area statistics based on supervised classification at the Brunei Bay.

Species	Area (ha)	Total	% contribution	
	Limbang	Sundar (up to Sipitang village)	Weston + Menumbok			
R. apiculata	782.45	5,003.25	7,016.19	12,801.89	36.38	
S. caseolaris	751.05	1,407.28	3,374.77	5,533.10	15.73	
S. alba	169.37	270.11	536.90	976.38	2.77	
N. fruticans	2,566.90	2,728.40	9,583.97	14,879.27	42.29	
X. granatum	741.65	197.42	54.03	993.10	2.83	
Total:	5,011.42	9,606.46	20,565.86	35,183.74	100.00	
Notes.

Genus names R Rhizophora

S Sonneratia

N Nypa

X Xylocarpus

Table 4 Mangrove structural estimates.

Mangrove tree structural estimates at Limbang, Sundar, Weston and Menumbok estuaries, Brunei Bay.

Estuary	Species	Density (trees 0.1 ha−1)	Basal area (m2 0.1 ha−1)	Relative density (%)	Relative dominance (%)	Relative frequency (%)	IV	Average tree height (m)	CI	
Limbang	Bruguiera gymnorrhiza	10	0.07	5.9	0.1	10.0	16.0	6.0	844	
	Nypa fruticans	38	20.40	23.5	17.0	30.0	70.5	6.8		
	Sonneratia caseolaris	77	51.41	47.1	42.8	40.0	129.9	11.7		
	Xylocarpus granatum	38	48.29	23.5	40.2	20.0	83.7	14.0		
	Total	163	120.17	100	100	100				
Sundar	Avicennia alba	6	0.29	4.3	0.9	6.5	11.7	11.0	303	
	N. fruticans	23	14.73	17.0	44.3	16.1	77.4	6.7		
	Rhizophora apiculata	54	5.25	40.4	15.8	25.8	82.0	13.0		
	S. alba	20	7.90	14.9	23.8	19.4	58.1	13.1		
	S. caseolaris	11	0.54	8.5	1.6	12.9	23.0	9.0		
	X. granatum	20	4.53	14.9	13.6	19.4	47.9	12.3		
	Total	134	33.24	100	100	100				
Weston	B. cylindrica	8	0.20	6.1	0.6	9.7	16.4	11.4	239	
	N. fruticans	31	25.91	24.2	73.8	29.0	127.0	7.7		
	R. apiculata	20	1.55	15.2	4.4	12.9	32.5	11.6		
	S. caseolaris	51	4.67	39.4	13.3	32.3	85.0	11.1		
	X. granatum	20	2.77	15.2	7.9	16.1	39.2	11.6		
	Total	130	35.10	100	100	100				
Menumbok	Ceriops sp.	2	0.01	1.5	0	2.9	4.4	9.0	519	
	N. fruticans	42	50.01	31.8	61.6	28.6	122.0	7.9		
	R. apiculata	74	30.56	56.1	37.6	51.4	145.1	14.9		
	X. granatum	14	0.59	10.6	0.7	17.1	28.4	11.6		
	Total	132	81.17	100	100	100				
Notes.

IV importance value

CI complexity index

The ground inventory also revealed no detectable changes in relation to the four-year old satellite (ALOS) data. The core mangroves were found intact in almost all locations between 2010 and 2014. The only sign of change was observed at Limbang, Sundar and Weston river mouths where shrubby vegetation (as understood from the visual interpretation of the ALOS data before the fieldwork) had become grown-up Sonneratia trees (height: 9–13 m). The missing mangrove patch (observed from the Google Earth) on the northern most corner of Menumbok was ca. 3,741 ha and found to be co-dominated by R. apiculata and N. fruticans (similar to other mangrove patches at Menumbok).

Figure 5 Mangrove diameter distribution.

Mangrove diameter class distribution at (A) Limbang, (B) Sundar, (C) Weston and, (D) Menumbok estuaries. Dotted box in (A–D) shows the range of diameter contributed by more than 50% of the trees.

Discussion

In recent years, remote sensing technology has greatly enhanced our understanding of mangrove ecosystems (Walcker, Gratiot & Anthony, 2016). The present study also found several interesting observations about the Malaysian mangrove cover at Brunei Bay. As of 1st September 2010 (ALOS data), the spatial extent of the mangroves was 35,183.74 ha (Fig. 2 and Table 3). However, if the missing mangrove patch of 3,741 ha is considered then the total mangrove cover should be ca. 38,924.74 ha. The ALOS data were efficient for mapping dominant mangrove species in the study area. Earlier, Hamdan, Khali Aziz & Mohd Hasmadi (2014) have used the ALOS data for identifying above ground biomass of the Matang Mangrove Forest Reserve (in Peninsular Malaysia) and found it advantageous to assess the vegetation across larger areas. The noise observed in the present supervised classification was chiefly associated with misclassification in-between R. apiculata, S. caseolaris and N. fruticans species. Differences in both quantity and exchange indicate that the area covered by N. fruticans was slightly underestimated due to R. apiculata and S. caseolaris interference in some locations (Figs. 6A–6B). Perhaps further understanding on the spectral reflectance properties of Nypa, Rhizophora and Sonnertia spp., would be able to improve the classification accuracy. In addition, application of the nonparametric algorithms like a Decision Tree could become advantageous for future mangrove mapping attempts (Zhang et al., 2017). The freely available moderate resolution satellite data (e.g., Sentinel) be a chance to correspond with the dates of ground inventory. On the other hand, pinpoint discrimination of the vegetation species, mixed with different age groups (juvenile, young and adult), is not practicable through the use of remote sensing data (Xie, Sha & Yu, 2008).

Figure 6 Fieldwork photographs.

Photographic evidences showing (A) co-dominance of Nypa fruticans and Rhizophora apiculata at Sundar (signboard indicates the existence of crocodiles in this area), (B) co-dominance of N. fruticans and Sonneratia caseolaris at Weston, (C) succession of S. caseolaris at Limbang river mouth, (D) mangroves facing surplus inundation of the high tide at Sundar, (E) uprooted mangrove trees along the border facing bay waters, (F) dense Rhizophora vegetation at Menumbok (photos taken by Behara Satyanarayana).

Mangrove species distribution at Brunei Bay (Fig. 2) coincides with the zonation patterns reported elsewhere. For example, succession by Sonneratia spp. at Limbang, Sundar and Weston river mouths indicates their pioneering nature along the open coasts on silty and silty-sand substrates (Fig. 6C) (Satyanarayana et al., 2002; FAO, 2007; Satyanarayana et al., 2010). However, S. caseolaris is also known to colonise the elevated (upstream) grounds (Saenger, 2002), and this could be a reason for its inland occurrence at Brunei Bay. According to Nik Nurizni (2015), the varying surface water salinity between 7.2–19.3‰  and 0.6–21.3‰  at the downstream locations of Limbang (depth: 1.8 m) and Sundar (depth: 2.5 m) shows a considerable freshwater discharge in both areas (no water quality measurements are available for Weston). In contrast, the lower abundance of Sonneratia spp. at Menumbok river mouth as well as upstream areas could be linked to the lack of sediment accretion grounds (Fig. 2) and deep water channels (depth: 4.6 m and salinity: 22.8–27.9‰) (Nik Nurizni, 2015). The species like Nypa fruticans which inhabit soft and fine-grained substrates in the areas of strong freshwater discharge and R. apiculata in the areas of strong neritic incursion (cf. Tomlinson, 1986; Saenger, 2002; Teh et al., 2008; Satyanarayana et al., 2010) have shown their widespread distribution at Brunei Bay, whereas X. granatum is confined to the interior and elevated grounds (cf. Satyanarayana et al., 2002; Satyanarayana et al., 2009).

Tree structural parameters obtained from the mangroves are not only useful to identify the vegetation status, but also for monitoring/management through the silvicultural practices (Dahdouh-Guebas & Koedam, 2006). Among others, basal area represents wood volume and is beneficial to assess any vegetation in terms of its succession or maturation (Satyanarayana et al., 2002; Satyanarayana et al., 2009; Satyanarayana et al., 2010). At Brunei Bay, the higher basal area at Limbang and Menumbok shows the more matured nature of the trees as opposed to Sundar and Weston (Table 4). While the majority of trees with a diameter of 31–90 cm have contributed to the highest basal area at Limbang, the trees with a 2.5–120 cm diameter were responsible for Menumbok. The diameter limited to 60 cm and less for a greater number of trees catered lower basal area for Sundar and Weston (Fig. 5). The complexity index, that depends largely on density, basal area and tree height estimates (Holdridge et al., 1971), could represent less/undisturbed nature of the vegetation along with its potential contribution to the biodiversity at both Limbang and Menumbok (cf. (Parkes, Newell & Cheal, 2003; Kovalenko, Thomaz & Warfe, 2012; Bartholomew, Hafezi & Karimi, 2016; Tongway & Ludwig, 2011). The geographic location of Sundar and Weston, which is rather under the direct influence of bay waters, must be accountable for its less structural complexity. The mangroves here seemed to be experiencing a stressful environment due to surplus inundation, flood and ebb water current, etc., which in turn causing some trees to uproot (Figs. 6D–6E). It should be noticed that mangrove establishment and growth attains better along sheltered coastlines than to the open areas (Alongi, 2008).

The differences in terms of important mangrove species at each estuary between remote sensing and vegetation survey are understandable as the supervised classification provide details of the entire mangrove cover (Table 3), whereas the PCQM provide details of the two transect-based observations (Table 4). In fact, both ground-truth and remote sensing results were virtually complementing each other at the places of vegetation inventory (Fig. 2). However, we draw the attention to the preliminary nature of our study, emphasizing the need for longer transects and more sample points in multiple homogeneous mangrove patches (cf. Engeman et al., 1994; Dahdouh-Guebas & Koedam, 2006) to estimate the forest structural parameters with higher accuracy. In view of recent literature it might also be more interesting to measure the second or third nearest tree when applying PCQM (cf. Hijbeek et al., 2013; Khan et al., 2016).

The results of this study however had a limitation of comparison with other mangrove studies at Brunei Bay due to poor scientific literature. Also, our efforts to find the actual (Brunei Bay) mangrove area from Sabah and Sarawak Forestry Departments did not yield satisfactory results. Though Sabah Forestry Department has informed the mangrove cover at Weston and Menumbok as 14,932 ha (SA Sani, pers. comm., 2016), but this figure was different from the present observation i.e., 24,306.86 ha (20,565.86 ha [ALOS-based] + 3,741 ha [Google-based]). The main reason for this difference was due to ‘state land’ and ‘forest land’ categories of mangrove vegetation at Brunei Bay (Forestry Department do not consider the state land forest). Therefore, the present study (with dendrometric measurements and species-level mapping) is able to serve as a strong baseline data for future mangrove investigations at Brunei Bay, including for monitoring and management purposes locally at present.

Conclusions

Malaysian mangrove cover at Brunei Bay, from Limbang, Sundar, Weston and Menumbok estuaries, was evaluated for the first time by means of scientific measures invoking remote sensing (ALOS) and ground-truth (PCQM) data. Moderate resolution of the ALOS data was efficient for mapping dominant mangrove species in the vicinity. The spatial extent of the dominant mangroves was estimated at ca. 35,183.74 ha where N. fruticans occupied the highest land-cover followed by R. apiculata, S. caseolaris, X. granatum and S. alba in the order. Also, ground inventory revealed the abundance of N. fruticans and R. apiculata as widespread species while S. caseolaris, X. granatum and S. alba as locally distributed species at the river mouth and/or interior/elevated grounds. The other less dominant species such as A. alba, B. gymnorrhiza, B. cylindrica and Ceriops sp. contributed insignificantly to the vegetation structure. High basal area at Limbang and Menumbok represents more matured nature of the vegetation as comparted to Sundar and Weston. Geographic location facing the direct influence of bay waters, surplus inundation, and flood and ebb water current were some of the issues believed to be responsible for low basal area and less mature vegetation at Sundar and Weston. Further understanding on the spectral reflectance properties of the co-dominant Nypa, Rhizophora and Sonnertia spp., along with application of the nonparametric algorithms like decision-tree, would be able to improve the accuracy of mangrove land-cover mapping. Overall, both remote sensing and ground-truth observations are in a general agreement to represent the (dominant) mangrove species composition and distribution at Brunei Bay and be able serve as a strong base-line data for future investigations.

Supplemental Information

File S1 Supplementary file—Limbang

Raw data for Limbang.

Click here for additional data file.

File S2 Supplementary file—Menumbok

Raw data for Menumbok.

Click here for additional data file.

File S3 Supplementary file—Sundar

Raw data for Sundar.

Click here for additional data file.

File S4 Supplementary file—Weston

Raw data for Weston.

Click here for additional data file.

The ALOS satellite data was provided by the JST under the Ocean Remote Sensing Project of the WESTPAC. The authors are grateful to administrative authorities at the JST and the UMT. Special thanks go to Mr. Subarjo and Mr. Kamal—the Science Officers at the INOS—for their kind support in the fieldwork. The courtesies extended by Mr. Ismail and his family at Lawas are greatly appreciated. Help rendered by the UMT undergraduate students—Nik Nurizni Nik Ali, Nooratikah Binti Mohd Din, Wajihah Binti Mat Nawi, and Rafidah Binti Abd Wahab—was invaluable.

Additional Information and Declarations

Competing Interests

Author Contributions

Field Study Permissions

Data Availability

The authors declare there are no competing interests.

Behara Satyanarayana conceived and designed the experiments, performed the experiments, analyzed the data, prepared figures and/or tables, authored or reviewed drafts of the paper, approved the final draft.

Aidy M. Muslim analyzed the data, contributed reagents/materials/analysis tools, approved the final draft.

Nurul Amira Izzaty Horsali and Nurul Ashikin Mat Zauki performed the experiments, analyzed the data, prepared figures and/or tables.

Viviana Otero analyzed the data, prepared figures and/or tables, authored or reviewed drafts of the paper, approved the final draft.

Muhammad Izuan Nadzri prepared figures and/or tables.

Sulong Ibrahim and Mohd-Lokman Husain conceived and designed the experiments, contributed reagents/materials/analysis tools, approved the final draft.

Farid Dahdouh-Guebas authored or reviewed drafts of the paper, approved the final draft.

The following information was supplied relating to field study approvals (i.e., approving body and any reference numbers):

This study was conducted with the permission of the State Forestry Department of Sarawak (# NCCD.907.4.4 (Jld. 10)-294).

The following information was supplied regarding data availability:

The raw data has been provided as Supplemental Files.

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
