# Peer review of "Status of the undisturbed mangroves at Brunei Bay, East Malaysia: a preliminary assessment based on remote sensing and ground-truth observations"

_PeerJ, doi:10.7717/peerj.4397_

## Round 0.1 · original submission · Major Revisions

The paper was reviewed by 3 reviewers, and most of them agreed that the topic is worth a publication. However the main concern is that the English is not in an acceptable form (Reviewer 1 & 3), and thus I highly recommend that the paper to be thoroughly re-written.

In addition, reviewer 3 is concerned about the analysis, i.e. more can be analysed such as diversity. In addition, the authors should also point out limitations in tropical areas (such as cloud cover) etc.

Reviewer 1 ·

Basic reporting

Unfortunately, the English quality is so poor as to render the paper unreadable in parts. I am only critical of English in scientific papers when it interferes with meaning or understanding, and the use of English in this paper makes sections difficult to comprehend. The first few sentences all have significant structural issues making them difficult to understand and much of your abstract better belongs in an introduction. In addition to the English problems and suitability for an abstract, the sentences are missing necessary information such as the year of measure yet contain superfluous information such as the joint administrative structure of Brunei Bay (again not abstract material). I highlight these problems as they are consistent throughout the paper. Below are the first few sentences and a proposed solution.

“Out of 709,700 ha total mangrove cover in whole Malaysia, East Malaysia (i.e. Sabah and Sarawak) is supporting as high as 596,867.90 ha whereas West (Peninsular) Malaysia only 112,832.10 ha. The mangrove vegetation at Brunei Bay is jointly shared as well as administered by East Malaysia and Brunei Darussalam. Four major estuaries namely, Limbang, Sundar, Weston and Menumbok are supporting the entire stretch of bay mangrove complex.”

How about..

As of 2012, Mangrove canopy cover in Malaysia was 709,700 ha with East Malaysia containing 596,868 ha of mangrove and West Malaysia 112,832 ha. The four estuaries of Limbang, Sundar, Weston, and Menumbok contain XX% of the total mangrove forest cover in Malaysia.


Others use Riverine, Basin, Fringe or sub-categories of these (see Ewel KC, Twilley R, Ong JE. Different Kinds of Mangrove Forests Provide Different Goods and Services. Global Ecology and Biogeography Letters 1998, 7: 83-94) for a good example. Nobody uses bay mangrove complex that I am aware of. I am not clear what a bay mangrove complex is, and I would advise against defining another type of mangrove from the traditional grouping.

Your abstract needs a complete rewrite. I made some notes and rewrote your first three sentences above as an example.

My advice is to hire an English speaking scientific writer.

The writing does improve after the abstract.

Experimental design

Lines 96 – 99, it is really water temperature that matters.

How did you ensure the accuracy of the topo-map?

Line 105 – How did you know it was mangrove, why manually digitized, why was this “better?” What digitizing protocols did you use. You are saying you manually digitized mangrove but how did you know it was mangrove?

Why a 4-year difference from RS collection to field validation? How may this cause issues?

Line 125 – How can a singular GCP be more than 100m2 in area?

Line 146 – Why 100m transects? Why these transect locations?


The methods need expanding to answer the questions above.

Validity of the findings

Line 211 – I would say RS is the standard now and has been for some time, it is not the alternate.

Due the questions I raise in your methods I cannot honestly say that the validity of your results are sound.

Additional comments

Line 60. This is not scientific writing, “luxuriant growth of pristine mangroves at Brunei Bay,”

Line 208. what is, "authentic mapping"?

The abstract assumes knowledge of the area to the village level. This should not be the case.

Line 55. These cites are quite dates and newer and improved estimates exist.

Line 62, I find this hard to believe, I know some work has been conducted on Malaysian mangroves.

Remove Lines 67 – 71.

The massively long paragraph around Line 216 goes off-topic.

The discussion is too long and off-topic.

Reviewer 2 ·

Basic reporting

1. In lines 95 to 97, how are the monsoonal regimes going to affect the mangroves in Brunei Bay? How does it alter the temperature and precipitation? Could you elaborate on the effects?
2. For line 151, if species’ Importance Value is a combined measure of relative density, relative dominance, and relative frequency, is it redundant to calculate it, or why is the combined measure better?
3. For line 152, could you please explain how Complexity Index is calculated?
4. For the ground inventory in general, could you include how these measurements are going to improve our understanding of the ecology of the mangroves or the impact on the estuaries?
5. In lines 187 to 193, you commented on the variation in basal area and density for mangroves in different estuaries. Do you have any explanation on the drivers behind these variations? For example, how does the water quality (dissolved oxygen, salinity, nutrient level) and flow dynamics differ in these estuaries?
6. In lines 271 to 273, could you explain why the influence of bay water will decrease basal area? Are the trees inundated with bay water?
7. For table 1, it might be helpful to provide the total mangrove cover for west coast, east coast, and East Malaysia for ease of comparison. It can also be beneficial to include a column of percent area to give the audience a better idea of the distribution of mangroves in your study area.
8. For table 3, it will be nice to have the percent area for each mangrove species.
9. The language is professional throughout.
10. The figures are very relevant, of high quality and clear to follow.
11. It might be good to include some information on the spectral profiles of the different species for future studies.

Experimental design

1. The research question is well defined, and the area is understudied in the literature.
2. The processing of the remote sensing data is standard. (Does the ALOS dataset come with the atmospheric correction?)
3. In line 108, could you be more explicit and elaborate on your supervised classification step? Did you train on four classes of dominant mangrove taxa and the unclassified, and how many samples did you train on for each class?
4. The use of GCP in the experimental design makes the result more rigorous. The ground data of dominant mangrove taxa provides good information on color and texture for the remote-sensing work.
5. The accuracy assessment with quantity, exchange, and shift is rigorous.
6. In line 139 to 140, how would you expect the spectral signature of mangrove to be affected with inconsistency in time between satellite image and fieldwork?

Validity of the findings

1. The findings from the ground-truth and remote sensing data are in agreement.
2. The data is robust, with both field and remote sensing data.
3. The subject of the study (mangroves at Brunei Bay) is novel and comprehensive.
4. The conclusion links back to the research question and clearly states the potential use of the result.

Additional comments

The manuscript can be improved to involve more scientific background on the different mangrove species and possible explanations of the different distributions of species among estuaries.

Reviewer 3 ·

Basic reporting

The English language should be improved to ensure that your international readers can clearly understand the information provided effortlessly. In particular, I was found numerous instances of sentences/ paragraphs starting with link words (e.g. despite, nevertheless, although, while) One option is requesting any of your native English speaking colleagues to review the manuscript. Some examples where the language could be improved include lines 22-23, 42-46, 139-144, 165-167, 176-180, 226-230, 276-280, 284-286

In the title a better choice would be ‘undisturbed’ instead of ‘pristine’.

Your introduction section needs more details on use of remote sensing technology for mangrove mapping. The material in discussion section is a better fit for introduction section. See my comments in discussion section. You should also expand upon the knowledge gap in both the study area and mangrove mapping using remote sensing.

L36-40: You are reporting estimates of multiple properties which is not always desirable in abstract. It will be helpful if you can simplify this reporting.

L40: The readers have no idea of the Complexity Index after reading first few lines of the abstract.

L49: It is not clear why “bay-mangrove complex, forest nature” are chosen as keywords.

Line 53: Sixth rank for what? In terms of area?

L66-67: Scientific attention for what purpose?

How is the statement in L194-196 is connected to the next (L196-204)

L319: ‘can serve’ not ‘can be severed’

L323: ALOS data not ALSO

I could not find the table and figure title which made it difficult to comment on those.

Table 2: No need to include both errors and accuracy -- reporting just the accurries would be sufficient.

Section 2.3: Field photographs would be useful for the readers.

Experimental design

The remote sensing component and ground inventory can be linked in various aspects. You have used only the location information for species mapping. With the rich ground inventory you have collected, I would strongly suggest to include a section on estimating the complexity/diversity using of mangrove species remote sensing data

Section 2.2: I find the remote sensing methodology section to be particularly weak. Why did you use ALOS data for this study? For e.g. freely available Sentinel 2A data has similar spatial resolution as ALOS data and could have provided up-to-date imagery for this research. Please provide a clear list of input bands for classification. Why did you use maximum likelihood classification (MLC) only? Fassnacht et al (2016) reported that nonparametric classifiers (decision trees, random forest, support vector machines) performs better than parametric methods like MLC for vegetation classification. (http://www.sciencedirect.com/science/article/pii/S0034425716303169)

I like the use of accuracy assessment following the method by Pontius and others, however more description about this method would be helpful for the readers.

L152: Please provide a description of the complexity index.

L153: In what context did you use P-DATA PRO software?

Validity of the findings

Section 3: Please divide the result section in relevant subsections with headings. E.g. 3.1 Classification accuracy of mangroves

L187-188: Same information has been provided in L176-178. You can merge the reporting and mention that similar observations were found in both remote sensing and ground observations.

How is the statement in L194-196 connected to the next (L196-204)?

Section 4:
L207-212: better fit for introduction section
L216-226: better fit for introduction section

L 224-226: I don’t agree completely. Mangroves are mostly seen in tropical areas where cloud is a big challenge for remote sensing community. To regularly map and monitor mangrove extents, SAR data could be as good as optical data if not better.

Additional comments

The manuscript reports current status of mangroves in Brunei Bay using Remote Sensing technology and extensive field observations. There is definitely a huge need for this kind of study to monitor and protect one of the vulnerable ecosystems of the world. However, the study designs are not adequate and manuscript needs language editing.

---

## Round 0.2 · Minor Revisions

The reviewer requires the authors to make a further minor revision.

Reviewer 3 ·

Basic reporting

Writing was among the biggest concern for this manuscript. However the current manuscript has improved a lot thanks to the extensive language editing.
Table 4: Please remove the black-background for the ‘total’ estimates. You can perhaps use font in bold for that row.
Figure 1: Scale needed for left hand inset; scale bar is not visible on the ALOS image.
Figure 4: Should the title be “mean spectral reflectance…” or “average spectral reflectance…”

Experimental design

I am not quite satisfied with the reason you provided justifying the use ALOS data. The time difference between ALOS data and field data collection is 4 years. For Sentinel-2, the time difference with field data would be 1 year only, which makes Senitnel-1 a better choice. Perhaps you can add this information in the section where you mention the possible issues with time gap.

I am also not fully convinced with the reason behind the selection of MLC over other non-parametric classifier. It is completely wrong to say that MLC is still considered as the benchmark for mangrove mapping related study based a single paper. You can cite the paper by Kuenzer et. al., 2011 (http://www.mdpi.com/2072-4292/3/5/878) to justify the choice of MLC. However, please remember that numerous studies in the last 6-7 years have used more advanced methods. Mangrove mapping is not an isolated problem of remote sensing image classification. While the majority of remote sensing community is moving towards a non-parametric image classification methods, it will unwise to not to adapt/test those methods for mapping mangroves.

Validity of the findings

None to comment

Additional comments

None to comment

---

## Round 0.3 · accepted · Accept

The authors have responded to all of the reviewers' comments. There are still few typos need to be checked in the final production, e.g. e.g. L. 56, airborne